# Physical Activity Barriers and Assets in Rural Appalachian Kentucky: A Mixed-Methods Study

**DOI:** 10.3390/ijerph18147646

**Published:** 2021-07-19

**Authors:** Natalie Jones, Deirdre Dlugonski, Rachel Gillespie, Emily DeWitt, Joann Lianekhammy, Stacey Slone, Kathryn M. Cardarelli

**Affiliations:** 1Family and Consumer Sciences Extension, College of Agriculture, Food and Environment, University of Kentucky, Lexington, KY 40506, USA; Natalie.jones@uky.edu (N.J.); Rachel.Gillespie@uky.edu (R.G.); emily.dewitt@uky.edu (E.D.); joann.lianekhammy@uky.edu (J.L.); 2Sports Medicine Research Institute, College of Health Sciences, University of Kentucky, Lexington, KY 40506, USA; dee.dlugonski@uky.edu; 3Department of Statistics, College of Arts and Sciences, University of Kentucky, Lexington, KY 40536, USA; stacey.slone@uky.edu; 4Department of Health, Behavior & Society, College of Public Health, University of Kentucky, Lexington, KY 40504, USA

**Keywords:** physical activity, rural, socioecological model, community-based participatory research

## Abstract

Obesity is an increasing public health concern in the U.S. and a contributor to chronic illness, with trends revealing a rise in adult obesity and chronic disease rates among the most vulnerable and disadvantaged populations, including those in rural communities. A mixed-methods approach was used to examine perspectives on perceived physical activity barriers, resources, and level of community support. Researchers utilized the socioecological model to examine the multiple domains that support physical activity in rural Appalachia. The present study focuses on baseline data, including a cohort survey to assess physical activity, health status, and barriers to physical activity, and five focus groups with elected community leaders, community residents, members, and key stakeholders to assess perspectives on physical activity barriers and resources within the county. The cohort survey sample (*N* = 152) reported a median of 6 barriers (range 0–13) to participating in at least 30 min of physical activity daily. The qualitative analysis yielded three overarching themes related to physical activity participation: lack of motivation, physical environment, and cultural barriers. This mixed-methods study revealed the challenges and perceptions among rural residents across the socioecological model when assessing physical inactivity. Findings can be used to tailor future interventions focused on expanding social support, designing infrastructure, and creating policies that promote physical activity.

## 1. Introduction

Present guidelines recommend that adults in the United States (U.S.) engage in at least 150 min of moderate-intensity or 75 min of vigorous-intensity physical activity a week [1]. Currently, 32.3% of Kentucky adults report no physical activity beyond their regular job in the past 30 days [2]. While the benefits of physical activity are widely documented, most adults in rural areas report participating in less physical activity than recommended [3]. In the central Appalachia region of the U.S., physical inactivity is even higher, with 33.8% of the population reporting no physical activity [4].

Obesity is an increasing public health concern in the U.S., and an important contributing factor to chronic disease, with trends revealing a rise in both adult obesity and chronic disease rates among the most vulnerable and disadvantaged populations [5], including those in rural communities [6]. The Appalachian region of the U.S. experiences a high obesity prevalence accompanied by numerous chronic diseases, including higher rates of heart disease and diabetes [7,8]. The magnitude of physical inactivity in Appalachia, and its effects on overall health, warrants detailed exploration to identify barriers related to physical activity participation among residents. 

The socioecological model (SEM) has been used widely for health promotion over the past three decades [9]. Figure 1 depicts the SEM and describes the multiple levels of influence on health behavior. The SEM can be used to identify relationships among multiple levels that influence rural residents’ physical activity behavior [9]. Previous research demonstrates the effectiveness of the SEM framework for understanding and guiding population-based health behavior interventions [10,11,12] and for focusing on encouraging physical activity [13].

The SEM suggests that the social and physical environments are important determinants of physical activity engagement [11]. In rural communities, activity-friendly community aspects and encouragement from support systems have aided in increasing physical activity engagement [14]. However, many barriers to physical activity still exist in rural areas. Individual-level barriers to activity include lack of motivation, lack of knowledge on how to participate, and lack of understanding of health impacts [15]. Community-level barriers are attributed to the physical environment, including lack of access to safe spaces for activity and sidewalks [16]. Policy implications lend opportunities to explore neighborhood, cultural, and social norms and the encompassing impact these can have on physical activity engagement within rural communities. Exploring these influential factors together provides an interpretation of the unique considerations required to understand and promote physical activity behavior.

The overall aim of this study is to gain perspectives from community residents and stakeholders about physical activity in a rural Appalachian setting to inform future intervention development. The objectives of this study are to (1) identify commonly reported barriers to participating in physical activity; (2) examine the relationship between self-reported barriers and physical activity participation; and (3) explore perspectives on perceived physical activity barriers, resources, and level of community concern about physical inactivity across the levels of the SEM.

## 2. Materials and Methods

### 2.1. Setting

This study took place in Martin County, located in eastern Kentucky, central Appalachia. Approximately 34.4% of residents live in poverty and the county struggles with low educational attainment and outmigration [17]. In 2019, 97.0% of residents utilized a car, truck, or van for transportation, while only 1.7% reported walking and 0.0% reported using a bike for transportation [18]. Additionally, 21.0% of the population, under age 65, reported living with a disability [17].

### 2.2. Study Design

A mixed-methods approach was used to examine our specific study aims as part of a larger five-year Centers for Disease Control and Prevention (CDC) High Obesity Program (HOP) project. This two-pronged design provided a complimentary approach toward accomplishing the objectives of this study. With cohort survey data, researchers were able to statistically assess common barriers to physical activity as well as explore its relationship to physical activity participation. The focus group data allowed for richer insight into residents’ perspectives on barriers and assets unique to their area and provided them the opportunity to voice community concerns related to physical inactivity.

The present study focuses on baseline data, including a cohort survey to assess physical activity, health status, and barriers to physical activity, and focus groups with elected community leaders, community residents, and key stakeholders to assess perspectives on physical activity barriers and resources within the county. The quantitative and qualitative data were collected consecutively. Cohort surveys were obtained in October and November of 2019 and focus group interviews were conducted in September and October of 2019. All study participants received a $25 incentive to a local grocery store for their participation. The Institutional Review Board (IRB) approved this study. The survey instrument and focus group moderator guide can be found in the Appendix A.

#### 2.2.1. Cohort Survey Recruitment

Martin County residents were invited to participate in a cohort study to longitudinally monitor changes in dietary intake, food accessibility resources and purchasing behaviors, community resources and perspectives, and physical activity behaviors. In order to be eligible for this study, participants had to be 21 years of age or older, have maintained residence in Martin County for at least one year, speak English, have no plan to move within the next three years, not be pregnant, and have never been diagnosed with cancer. Participants were recruited through the Martin County Cooperative Extension Service (CES) Office, the CES Facebook page, a local food pantry, several faith-based organizations, and two local grocery stores. DeWitt at el. outlines the recruitment and enrollment process for the cohort study [19].

#### 2.2.2. Cohort Survey Measures

The survey instrument was comprised of a variety of assessment tools including the National Cancer Institute (NCI) Fruit and Vegetable Intake Screener to assess fruit and vegetable intake, household environmental measures, and food outlet purchasing practices. The physical activity portion of the survey utilized questions from the Global Physical Activity Questionnaire (GPAQ) [20]. However, the aims of this manuscript will focus on two specific responses among study participants, in which the physical activity level variable was created from self-reported responses to the GPAQ. Participants were asked “In a typical week, do you do any vigorous-intensity sports, fitness, or recreational activities that cause large increases in breathing or heart rate for at least 10 min continuously? Examples include running, hiking, shoveling, or playing basketball.” Frequency was assessed for participants answering “yes.” The second question asked respondents a similar question, except for “moderate-intensity” physical activity engagement and noted, “Examples include brisk walking, bicycling, gardening or heavy cleaning.” Participants indicated yes or no, and then provided the number of days per week they engaged in moderate-intensity activities. These assessments were used to group participants into the overall physical activity categories: 0 days per week = ‘Inactive’; 1–4 days per week = ‘Moderately Active’; and 5–7 days per week = ‘Active.’ Barriers to physical activity were measured from questions taken from two instruments, the Influences on Physical Activity Instrument (Cronbach’s alphas for subscales = 0.53 − 0.77) [21] and the Physical Activity Barriers questionnaire (Cronbach’s alpha = 0.859) [22].

#### 2.2.3. Cohort Survey Statistical Analysis

All data were analyzed using SAS 9.4 (SAS Institute, Cary, NC, USA). Comparisons between the three physical activity groups were calculated using an analysis of variance for mean age and the Kruskal–Wallis test for the median number of barriers. Chi-square or Fisher’s Exact tests, as appropriate, were used to determine the association between activity level and the categorical variables (gender, race, education, income level and all barriers).

#### 2.2.4. Focus Group Recruitment

Martin County Extension agents and Community Coalition members purposively recruited participants for focus group participation. Informational flyers, approved by the IRB, were distributed via the Extension Office and the Martin County CES Facebook page. The inclusion criteria used to determine eligibility for the cohort survey study also applied to those interested in participating in the focus groups.

A trained moderator (K.C.) facilitated the focus groups, accompanied by two note takers (E.D. and R.G.). All focus groups took place in the Martin County Extension Office or the local middle school and lasted approximately one hour each. Cardarelli at el. provide additional details of the focus groups conducted in fall 2019 [23].

#### 2.2.5. Focus Group Analysis

Focus groups were audio recorded and transcribed verbatim using a grounded theory approach and aimed to gain a better understanding of resources available for making healthy choices in the community, the need for additional resources to promote healthy choices, and the barriers to facilitating factors to support healthy eating habits and physical activity in the community. This iterative qualitative analysis involves cycles of data coding, the inductive creation and revision of categories, repeated comparison of data to extant literature to initial conclusions and back to the data, collection and categorization of additional data, and restructuring of categories and conclusions as warranted. Researchers independently identified themes from the transcripts and field notes and analyzed the qualitative data with NVivo software version 12 (QSR International, Burlington, MA, USA). Before coding the data, researchers engaged in provisional coding and evaluating field notes to develop a consensus codebook that defined broad constructs of interest. During secondary coding, investigators engaged in constant comparative analysis, relating transcript content to both the predefined categories, as well as emergent themes. The research team met regularly to discuss, refine, accept, or reject newly suggested codes as well as to triangulate findings toward better understanding assets and structural barriers to physical activity.

## 3. Results

Sociodemographic data for cohort and focus group participants can be found in Table 1. The mean age of participants in both groups was between 50 and 55 years, with the majority identifying as female. Like the racial distribution of the county, almost all participants in both samples identified as white. Focus group participants had higher levels of education compared to those in the cohort study, with 85% of participants reported to have at least some college education. Cohort participants most frequently completed 11th grade or high school; over 43.4% of cohort participants were not high school graduates. Reported household income reflected a similar pattern among focus group participants, with 76% reporting an annual household income of $21,000 and above, whereas 60% of cohort study participants reported income less than $20,000.

Among the cohort sample, 31.6% (*n* = 48) were categorized as ‘Inactive’ and more than two-thirds of participants reported engaging in weekly physical activity. The majority, 44.1% (*n* = 67), were categorized as ‘Moderately Active,’ while 24.3% (*n* = 37) of the cohort study participants were categorized as ‘Active’ participants. No significant differences in the demographics were noted between the three activity levels. Overall, participants reported a median of 6 barriers (range = 0–13) to participating in at least 30 min of physical activity daily. The relationship between physical activity barriers and self-reported activity levels were explored. A statistically significant association (*p* < 0.001) was found between the number of barriers and activity levels, with the median of 4 (0–11) barriers for inactive participants being less than those among the moderately active and active groups (7 (1–13), 5 (1–12), respectively).

The qualitative analysis yielded three overarching themes related to physical activity participation: lack of motivation, physical environment, and cultural barriers. Themes and subthemes identified from the quantitative and qualitative analysis are shared under each corresponding level of the SEM, as previously described in Figure 1.

### 3.1. Individual Factors

Quantitative Findings. There was a relationship between participant activity level and making small changes to be more active, such as taking the stairs instead of the elevator (*p* = 0.002). Most participants (62.5%) in the inactive group responded ‘Never or Rarely,’ while 26.9% of the moderately active participants, and 29.7% of the active participants, never or rarely made small changes to be active. Over half of the participants in both the moderately active and active groups (59.7% and 51.4%, respectively) said they sometimes or often make small changes to increase activity. No significant differences in participant activity level, nor making small changes to be more active, were seen across education levels (*p* = 0.92 and 0.68, respectively).

Table 2 details frequencies and percentages for individual barriers by cohort participant physical activity level. Seven of the reported 13 barriers were found to be significantly different between activity levels. Access to proper clothing or shoes for activity as a barrier varied by activity level (*p* = 0.035), with 45.5% of moderately active participants reporting it as a barrier. Less than one-third of inactive (28.3%) and active (22.2%) participants reported it as a barrier. Lack of time as a barrier was another factor associated with activity level (*p* = 0.004). Most moderately active (66.7%) and active participants (56.8%) found that they did not have time to be physically active daily. Less than half of inactive participants (34.8%) reported time as an issue.

Participant physical activity level was also related to the presence of a health condition (such as asthma, COPD, or arthritis) (*p* = 0.046) as a barrier. A large percentage of moderately active (60.6%) and active participants (50.0%) identified a health condition as a barrier. Over three-fourths of inactive participants (76.1%) had a health condition that made it more difficult to be physically active regularly.

Qualitative Findings. Focus groups primarily focused on community-level factors related to physical activity participation. However, participants identified lack of motivation as a key personal obstacle to being physically active. This theme was described as the belief that a person is inactive because they have no desire to improve their health.

One participant said, “I would say probably 10% or less are physically active in our community.” Another participant highlighted the variability in physical activity among community members,

“I think it just depends, I think some people live a very active lifestyle and then some I think it’s just completely a toss-up. You know, some people are never active, but you know I think there’s just a divide. I think that, I don’t think we have a fully active community.”

Some participants thought the lack of motivation for physical activity and other healthy behaviors began early in life and with a younger generation that was focused on “iPads and air conditioning.”

### 3.2. Social Determinants

Quantitative Findings. Reliable childcare for engaging in physical activity was the only social barrier to physical activity. This determinant was reported less frequently—(15.5% among all physical activity groups)—than barriers in other levels of the SEM and there were no significant differences between activity levels.

Qualitative Findings. The social determinants included both social norms that exist in the community as well as social support for physical activity. One barrier subtheme encapsulated the idea that the culture of the region was not conducive to physical activity. Participants described their own community as having a history of being unhealthy, where leading a physically active lifestyle was not “the norm.”

“It’s never really been a part of our culture around here, it just isn’t. I mean we’re like the most unhealthy people in the country. This part, I mean that’s just honest, central Appalachia it is.”

The second subtheme was social support, particularly the importance of others encouraging participation. One participant shared how accountability from a friend motivated her to be active,

“My mom and I for example, and mom’s like “I don’t really want to walk today” and I am like ok no big deal. But if Sarah called me and said, “Hey you are supposed to walk with us ` today at 6 o’clock”. I am like oh gosh, can’t let Sarah down. We got to get off your butt let’s go. You know what I am saying? It is that unknown person that would come in and motivate some people because like my mom won’t do anything, I ask her to do anyways.” 

### 3.3. Physical Environment

Quantitative Findings. Barriers to physical activity in the physical environment are presented in Table 2 by cohort participant activity level. Access to safe places to walk (*p* < 0.001) was a barrier to physical activity mostly among moderately active participants (62.1%). The percentage of participants who expressed safety was a barrier was much lower in the inactive (30.4%) and active groups (22.2%).

Lack of space (e.g., home, living room, and backyard) as a barrier was associated with activity levels (*p* = 0.049). Over half of moderately active participants (51.5%) cited this as a barrier, compared to 41.7% of active and 28.3% of inactive participants. Access to facilities or space to be active (e.g., gyms, recreation centers, and green space) showed a similar trend (*p* = 0.001) in that majority of moderately active participants (57.6%) thought access to facilities or space was an obstacle. However, fewer active participants (27.8%) found this barrier a problem compared to their response regarding lack of space. The percentage of inactive participants who said the lack of access to facilities or space influenced the level of their daily physical activity remained the same (28.3%).

Most participants agreed that weather was a barrier to physical activity (*p* = 0.048). Active (77.8%) and moderately active (75.8%) participants more often cited this as a barrier than inactive (56.6%) participants.

Qualitative Findings. Participants provided many examples of how the physical environmental barriers contributed to a lack of physical activity, including those related to transportation, safety concerns, and inaccessibility of spaces to be active. One of the commonly discussed physical environment barriers was transportation, a challenge typically characteristic of rural areas. The distance and road quality between areas of the community and recreation facilities create a barrier for accessing physical activity spaces. One participant shared:

“No, I wouldn’t drive over here to walk on a trail cause it’s 30 min for me to get here.”

“You have hollers everywhere that people can’t, they have no way to get here [to town with recreation facilities].”

Other participants shared several different safety concerns about using existing public physical activity spaces. These concerns included fear of wildlife such as bears, cougars, and snakes on the trails, the presence of drug activity in parks, and dangers related to walking or biking on roads that were not designed to support active transportation.

“When I walk up there to the tower, I always have a gun, because there has been bears spotted up in there you know and all.”

“(Referring to a local park) They are finding a lot of needles over there. Cause the last birthday party we had over there, people was afraid to turn their kids loose.”

“We are a lot of back roads. You know, there’s not sidewalks everywhere.”

Participants described the inability to access existing physical activity resources as another barrier subtheme. For example, they shared that community centers are only open for walking at certain times during the winter. Additionally, participants noted that a track at a local school would be an ideal place for community members to walk, but it stays locked. Finally, participants noted that some walking trails were not handicap accessible.

“I mean anyone with knee trouble, back trouble, disabilities would have trouble. You couldn’t do it with a wheelchair you would have to stay on the road.”

### 3.4. Policy Determinants

Qualitative Findings. Participants recognized that their county possesses valuable assets and community resources for encouraging physical activity, but also noted opportunities for improvement. Participants viewed their community as a natural and beautiful place, optimal for outdoor physical activity pursuits. Participants expressed a desire for investment in designing the physical environment in a way that would enhance community health and enhance tourism.

“We need more parks, the parks we have need updated equipment, that sort of thing. Or livability type issues, you need to be able to safely walk a stroller around town and have someplace to go.”

“There’s trails, I mean we have one of the most beautiful places in the world to get out and do stuff.”

“I think that there’s a lot more grassroots stuff happening here now like we’re taking our communities back and we’re fixing it ourselves instead of people coming in and trying to fix us like they’ve always done. And it’s a power of the people and I think that’s what it is.”

## 4. Discussion

The goal of this study was to identify perceived physical activity barriers and resources across levels of the SEM in one rural Appalachian community in order to inform intervention development. Our data showed low levels of physical activity among residents, with 32% of participants reporting no physical activity at all. This is consistent with previous research indicating 35% of Martin County, and 33.8% of Central Appalachian adults, reported no physical activity or exercise in leisure time [24,25]. However, these findings present key insights as to the variety of barriers residents in this specific community experience, thus hindering their ability or desire to engage in physical activity. By conceptually applying these findings across the SEM, it provides an opportunity for the implementation of public health improvements across the different levels to address and improve physical activity levels through a multifaceted approach, which may elicit the greatest impact.

The most frequently reported individual-level barriers with significant influence on physical activity among our participants was the presence of a health condition. Although this trend was prevalent among all subgroups, inactive participants reported this barrier more frequently than moderately active or active participants. Further, a lack of time, lack of space, access to safe places to walk, and the weather were significant barriers to physical activity within the cohort. Moderately active or active participants cited these barriers more frequently than those who were inactive in our sample. While not significantly different by activity level, a lack of energy or motivation was the most frequently reported barrier to physical activity among all cohort participants. This was consistent with a key theme from the focus groups, in which a lack of motivation was identified as a primary obstacle to physical activity in this community. There are at least three plausible explanations for these patterns: (1) participation is difficult within an environment and culture that does not encourage or safely support physical activity, (2) being active increases awareness of barriers to physical activity participation, and (3) one’s own limitations due to health conditions may prevent individuals from engaging in activity. This indicates multicomponent interventions to address individual-level and physical environmental factors are needed to reduce barriers to physical activity and increase activity levels. Additional research is needed to explore strategies to modify behavior that must consider how to leverage interest and assets to improve engagement within this rural population.

Interestingly, the inactive group of survey respondents posed two clear barriers, lack of energy or motivation and presence of a health condition. This presents a unique cognitive dilemma, and considerable opportunity, for future research exploring what hinders physical activity engagement among individuals who are not active in any capacity within this rural Appalachian county. There may be a reciprocal relationship between lack of energy or motivation and health conditions that reinforces physical inactivity. However, more investigation is needed to probe this finding as it relates to the extent of environmental influence, cultural and social norms, or socioeconomic contributors. This highlights the potential need to create interventions that capture the interest of individuals who are currently inactive, particularly among this population. Although qualitative findings were unable to discern activity levels among participants, lack of motivation or energy was commonly reported. However, focus group questions probed participants and conversations on community resources and barriers to physical activity, which may explain why fewer variables at the individual level were identified.

The current study adds to the literature by revealing numerous barriers that may influence physical activity participation in rural Appalachia, and communities that are similar. The persistent disparities rural Appalachian communities face in being physically active, compared to urban and suburban residents, suggest that findings have not been effectively integrated into practice. Additionally, results from this mixed-methods study indicate that substantial behavioral and infrastructural improvements must be made in order to increase physical activity engagement in these communities. While several individual-level barriers were outlined in this study, barriers in the physical environmental and social barriers among this sample must be addressed before behavior change occurs. Kruger and colleagues [26] outlined the importance of appealing to cultural norms of rural Appalachia when designing interventions. This finding was echoed in this study as a consideration at multiple levels of the SEM and should be prioritized when designing tailored interventions to encourage physical activity in the region. Further, adults in this study identified several self-reported barriers to activity, yet there is a large group of moderately active individuals who, based on our data, are already taking small steps to be more active. This outcome indicates that the population may respond well to environmental and/policy changes that make it easier to be active in their community, posing a unique opportunity for future investigation.

The individual sphere of influence, most directly impacting one’s conscious decision-making, may deter the efficacy of other SEM spheres of influence to increase physical activity levels. However, other levels within the SEM are important points of intersection as viable interventions such as policies to enhance the health of the community through the physical environment. Participants in the present study desired park enhancements and trails that are safe to use. However, there are many challenges to environmental and policy changes in rural communities, including small population size, insufficient funding, and inadequate support from community leaders [27]. Making policy and environmental changes would not only increase access to these spaces, but increase the opportunity to influence change at multiple levels of the SEM to improve individual health behaviors and community health outcomes.

Utilizing the SEM to guide physical activity interventions in rural communities will help increase adherence to the multiple levels and systems that influence behavior. Research reveals the myriad factors impacting an individual across the SEM in rural communities [28], emphasizing that direct education programs alone are not sufficient in creating lasting behavior change. Therefore, the guidance of the SEM facilitates a multipronged approach to improve physical activity levels in rural communities, while Policy, Systems, and Environmental (PSE) interventions have shown to be effective approaches to improving health-related behaviors through a top-down approach in relation to the SEM [29]. Incorporating PSE efforts into future interventions could provide a micro-lens approach on a macro scale in Appalachian communities.

Limitations. Due to the cross-sectional nature of our cohort survey, our study design precludes causal inference. In addition, our sample was predominantly female among cohort and focus group participants. Focus group participants reported higher levels of education than the general population. Focus group participants were recruited via a purposive community-engaged approach, whereas cohort recruitment was comprised of a convenience sample. Due to this, our participants might have been more aware of where to find access to facilities or space to be active in their county. Therefore, generalizability may be limited. Finally, we cannot rule out errors in measurement of self-reported physical activity, such as over reporting due to social desirability or recall bias. However, such bias would indicate that we have overestimated levels of physical activity in this population.

## 5. Conclusions

This mixed-methods study revealed factors that are associated with physical activity engagement in a rural Appalachian county across various levels of the SEM. These findings can be used to guide physical activity research and help tailor public health interventions that focus on rural communities. The barriers and resources identified across the SEM can be used to guide the development of multifaceted interventions to reduce the challenges to being physically active and increase community supports and resources for promoting physical activity. Results of this study can inform future work that focuses on expanding social support in the community, designing infrastructure that supports physical activity, and creating policies that influence health to target an individual’s current level of physical activity.

## Figures and Tables

**Figure 1 ijerph-18-07646-f001:**
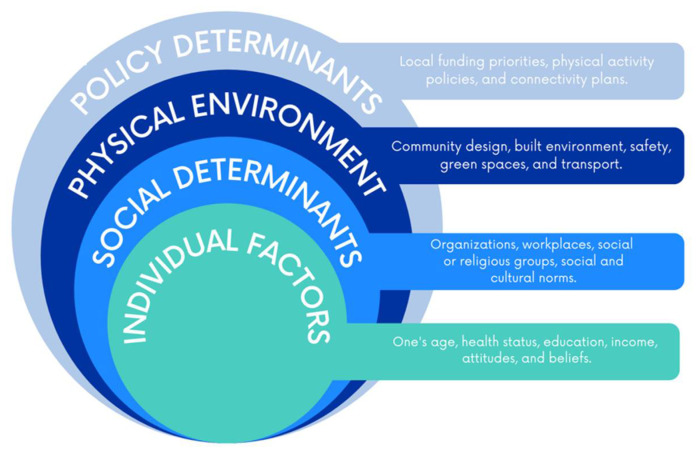
A socioecological model describing the multiple levels of influence for physical activity behaviors.

**Table 1 ijerph-18-07646-t001:** Sociodemographic characteristics of cohort study (*N* = 152) and focus group (*N* = 34) participants.

Characteristic	Cohort Study (*N* = 152)	Focus Group (*N* = 34)
Age, *M* (*SD*)	54.7 (15.3)	50.4 (13.1) years
Gender, *n* (%)		
Female	99 (65.1%)	27 (79%)
Male	53 (34.9%)	7 (21%)
Race, *n* (%)		
White	150 (98.7%)	34 (100%)
Education, *n* (%)		
11th grade or less	66 (43.4%)	1 (3%)
High school graduate	55 (36.2%)	4 (12%)
Some college	17 (11.2%)	12 (35%)
College graduate	14 (9.2%)	17 (50%)
Household Income, *n* (%)		
Less than $20,000	90 (60.4%)	8 (23%)
$21,000–59,999	44 (29.5%)	13 (38%)
$60,000 and above	15 (10.1%)	13 (38%)

**Table 2 ijerph-18-07646-t002:** Cohort Survey Respondents’ Reported Frequency of Barriers by Activity Level (*N* = 152).

	Inactive (*n* = 48)	Mod. Active (*n* = 67)	Active (*n* = 37)	All (*n* = 152)	*p*-Value
Lack of time **	16 (34.8%)	44 (66.7%)	21 (56.8%)	81 (54.4%)	0.004
Lack of energy or motivation	36 (80.0%)	58 (87.9%)	26 (70.3%)	120 (81.1%)	0.089
Lack of space *	13 (28.3%)	34 (51.5%)	15 (41.7%)	62 (41.9%)	0.049
Access to reliable childcare	7 (15.2%)	10 (15.2%)	6 (16.7%)	23 (15.5%)	0.977
Access to facilities or space to be active **	13 (28.3%)	38 (57.6%)	10 (27.8%)	61 (41.2%)	0.001
Access to proper clothing or shoes for activity *	13 (28.3%)	30 (45.5%)	8 (22.2%)	51 (34.5%)	0.035
Access to safe places to walk ***	14 (30.4%)	41 (62.1%)	8 (22.2%)	63 (42.6%)	<0.001
Cost	11 (23.9%)	27 (40.9%)	8 (22.2%)	46 (31.1%)	0.067
Weather *	26 (56.5%)	50 (75.8%)	28 (77.8%)	104 (70.3%)	0.048
Self-conscious	17 (37.0%)	34 (51.5%)	15 (41.7%)	66 (44.6%)	0.288
Health condition (such as asthma, COPD, or arthritis) *	35 (76.1%)	40 (60.6%)	18 (50.0%)	93 (62.8%)	0.046
Injury (such as a broken bone, recovery from surgery)	22 (47.8%)	27 (40.9%)	15 (41.7%)	64 (43.2%)	0.750
Lack of self-discipline	23 (50.0%)	41 (63.1%)	20 (55.6%)	84 (57.1%)	0.381

* *p* < 0.05; ** *p* < 0.01; *** *p* < 0.001.

## Data Availability

The data presented in this study are available on reasonable request from the corresponding author.

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
