# Peer review of "Physical Activity Barriers and Assets in Rural Appalachian Kentucky: A Mixed-Methods Study"

_ijerph, 2021, doi:10.3390/ijerph18147646_

Round 1

Reviewer 1 Report

A good paper with timely information for a problem long overdue. While the knowledge shared is not so new or innovative, the research provides a springboard for action.

Line 62 - As part of the second paragraph, talk about the Institutional/policy factors that will be addressed in the paper. since you adopted SEM in whole.

Lines 93 - 95 - Include the questions for the reader to know what was being measured and how these were captured in the instrument.

Lines 115 - 118 - Results showed that focus group participants had higher levels of education compared to those in the cohort study. The possible explanation for this finding is not given in the discussion (and no other part of the paper provides information that could help the reader gauge why this is so). Elaborate on this. Otherwise it appears simply as a selection bias.

Lines 192 - 194 - How does weather affect the participants? Too hot or too cold? This is not explained. 

The findings have not shown or explained how 'education level', and how physical activity knowledge (or lack thereof) influences level of physical activity yet these were part of the data collected from participants. Elaborate on these aspects.

Author Response

Reviewer 1

A good paper with timely information for a problem long overdue. While the knowledge shared is not so new or innovative, the research provides a springboard for action.

  1. Line 62 - As part of the second paragraph, talk about the Institutional/policy factors that will be addressed in the paper since you adopted SEM in whole.

Response: We have included a sentence indicating the policy and institutional opportunities that will be explored within the manuscript as it aligns with the SEM. This includes the cultural and social norms within this community that may influence physical activity among the population.

  1. Lines 93 - 95 - Include the questions for the reader to know what was being measured and how these were captured in the instrument.

Response: Additional details were added to report the specific GPAQ questions that were asked cohort survey participants to self-report. These questions included engagement and frequency of vigorous intensity and moderate intensity activities in order to quantitatively analyze our study sample’s physical activity levels. The GPAQ questionnaire is a validated assessment and scoring tool (Cleland, C. et al, 2014) to capture and assess physical activity levels and thus was utilized to measure current PA engagement among the study sample.

  1. Lines 115 - 118 - Results showed that focus group participants had higher levels of education compared to those in the cohort study. The possible explanation for this finding is not given in the discussion (and no other part of the paper provides information that could help the reader gauge why this is so). Elaborate on this. Otherwise, it appears simply as a selection bias.

Response: We have revised our limitations to include an explanation for this finding.
“Focus group participants were recruited via a purposive community-engaged approach whereas cohort recruitment was comprised of a convenience sample.”

  1. Lines 192 - 194 - How does weather affect the participants? Too hot or too cold? This is not explained. 

Response: The barriers questionnaire only asked participants to rate the extent to which “weather” prevents them from being active, with responses ranging from ‘none’ to ‘a great deal’. Unfortunately, this questionnaire does not provide any information about the ways in which weather influences participant physical activity level.

  1. The findings have not shown or explained how 'education level', and how physical activity knowledge (or lack thereof) influences level of physical activity yet these were part of the data collected from participants. Elaborate on these aspects.

Response: We included analyses comparing differences in levels of physical activity and small changes made to be more active across education levels. Level of education does not appear to influence activity level or frequency of attempts to make small changes to be more active.  

In summary, we appreciate the thoughtful comments of the two reviewers and the Editor and believe that the revisions reflect a much stronger manuscript. Thank you for considering this resubmitted manuscript, and please let us know if any of our edits require further clarification.

Sincerely,

Kathryn Cardarelli on behalf of all authors

Reviewer 2 Report

The theoretical basis is well founded and justifies the study. The objective is clearly defined. However, I believe that there is a need to adjust some items related to manuscript text:

(1) Abstract:

Lines 14-16 - I suggest justifying the study by increasing the prevalence of chronic diseases in vulnerable and disadvantaged populations, not just obesity.

Line 18 - Replace SEM with "social-ecological model"

(2) Introduction:

Lines 39-44 - I suggest expanding the justification for the study through the relationship between physical inactivity and chronic diseases among the most vulnerable and disadvantaged populations and rural communities.

(3) Results:

To facilitate data visualization, quantitative findings equivalent to individual factors (lines 139-158) and physical environment (179-194) should also be presented in tables.

Author Response

Reviewer 2

The theoretical basis is well founded and justifies the study. The objective is clearly defined. However, I believe that there is a need to adjust some items related to manuscript text.

Abstract:

  1. Lines 14-16 - I suggest justifying the study by increasing the prevalence of chronic diseases in vulnerable and disadvantaged populations, not just obesity.

Response: Thank you for this suggestion. We have added this to the abstract.

  1. Line 18 - Replace SEM with "social-ecological model"

Response: We have made this replacement.

Introduction:

  1. Lines 39-44 - I suggest expanding the justification for the study through the relationship between physical inactivity and chronic diseases among the most vulnerable and disadvantaged populations and rural communities.

Response: We have added this to the justification in the introduction section.

Results:

  1. To facilitate data visualization, quantitative findings equivalent to individual factors (lines 139-158) and physical environment (179-194) should also be presented in tables.

Response: Findings of individual factors related to the frequency of small changes for activity are presented but not shown in table format as it is the only question of its type. All other individual factors and physical environment barriers found in this section are presented in Table 2. We moved a sentence to the start of the paragraph to reflect this for section 3.1.1.  Table 2 details frequencies and percentages for individual barriers by cohort participant physical activity level. We added a sentence to the beginning of section 3.3 to reflect this presentation as well.  Barriers to physical activity in the physical environment are presented in Table 2 by cohort participant activity level.

Other minor changes were incorporated in the results section of the manuscript, including report of more specific p-values from Table 2. Minor grammatical improvements were also made to strengthen the readability of the manuscript. All changes are reflected with the track changes feature.

We also addressed comments embedded in the manuscript from MDPI. We added the specified departmental information for all authors and collapsed affiliations under one number for respective authors.

In summary, we appreciate the thoughtful comments of the two reviewers and the Editor and believe that the revisions reflect a much stronger manuscript. Thank you for considering this resubmitted manuscript, and please let us know if any of our edits require further clarification.

Sincerely,

Kathryn Cardarelli on behalf of all authors